# Multiparametric MRI followed by targeted prostate biopsy for men with suspected prostate cancer: a clinical decision analysis

Sarah R Willis,[1] Hashim U Ahmed,[2,3] Caroline M Moore,[2,3] Ian Donaldson,[2,3] Mark Emberton,[2,3] Alec H Miners,[1] Jan van der Meulen[1]

[1]Department of Health Services Research and Policy, London School of Hygiene & Tropical Medicine, London, UK
[2]Division of Surgery and Interventional Science, University College London, London, UK
[3]Department of Urology, University College London Hospitals NHS Foundation Trust, London, UK

Correspondence to
Sarah R Willis;
sarah.willis@lshtm.ac.uk

## ABSTRACT

**Objective:** To compare the diagnostic outcomes of the current approach of transrectal ultrasound (TRUS)-guided biopsy in men with suspected prostate cancer to an alternative approach using multiparametric MRI (mpMRI), followed by MRI-targeted biopsy if positive.

**Design:** Clinical decision analysis was used to synthesise data from recently emerging evidence in a format that is relevant for clinical decision making.

**Population:** A hypothetical cohort of 1000 men with suspected prostate cancer.

**Interventions:** mpMRI and, if positive, MRI-targeted biopsy compared with TRUS-guided biopsy in all men.

**Outcome measures:** We report the number of men expected to undergo a biopsy as well as the numbers of correctly identified patients with or without prostate cancer. A probabilistic sensitivity analysis was carried out using Monte Carlo simulation to explore the impact of statistical uncertainty in the diagnostic parameters.

**Results:** In 1000 men, mpMRI followed by MRI-targeted biopsy 'clinically dominates' TRUS-guided biopsy as it results in fewer expected biopsies (600 vs 1000), more men being correctly identified as having clinically significant cancer (320 vs 250), and fewer men being falsely identified (20 vs 50). The mpMRI-based strategy dominated TRUS-guided biopsy in 86% of the simulations in the probabilistic sensitivity analysis.

**Conclusions:** Our analysis suggests that mpMRI followed by MRI-targeted biopsy is likely to result in fewer and better biopsies than TRUS-guided biopsy. Future research in prostate cancer should focus on providing precise estimates of key diagnostic parameters.

## Strengths and limitations of this study

- There are no clinical studies that directly compare the standard diagnostic approach using transrectal ultrasound (TRUS)-guided biopsy in all men with suspected prostate cancer with an approach where multiparametric MRI (mpMRI) is used to select men for biopsy and to guide the biopsy needle towards a suspicious lesion against an accepted gold standard.
- Our decision analysis brings together emerging evidence on the diagnostic accuracy of TRUS-guided biopsy, mpMRI and MRI-targeted biopsies.
- A probabilistic sensitivity analysis demonstrates that the mpMRI-based strategy was most effective in 86% of the simulations. However this sensitivity analysis did not assess the impact of structural uncertainties.
- This analysis focuses purely on short-term clinical outcomes following different testing options. Ultimately, the optimal diagnostic strategy for men with suspected prostate cancer will depend on the impact on both costs and quality-adjusted life expectancy.

## INTRODUCTION

Prostate cancer is the most common male cancer in most developed countries. Incidence rates have risen rapidly over the past 15 years, in part due to the increase in prostate-specific antigen (PSA) testing. The use of PSA testing remains controversial as it lacks both sensitivity and specificity for the detection of prostate cancer.[1][2] Despite the high incidence, many men diagnosed with prostate cancer will not die from the disease so it is accepted that a distinction should be made between prostate cancer that is unlikely to cause harm ('clinically insignificant' disease) and cancer which, if untreated, may negatively impact quality of life or lead to death ('clinically significant' disease). While there is currently no agreed threshold of significance, most commentators agree that clinically significant disease should be declared when disease exceeds a certain volume or is populated by histological patterns that exhibit poor differentiation (Gleason score).[3–5]

The optimal strategy for diagnosing clinically significant prostate cancer is the focus of a rapidly developing body of research. The

standard diagnostic approach for men with suspected prostate cancer is to offer them a transrectal ultrasound (TRUS)-guided prostate biopsy taking 10–12 cores.[6–8] The ultrasound guidance ensures that the biopsy needles are guided to zones within the gland which are considered to have an equal probability of harbouring disease. An alternative to this is to identify areas of the prostate that are more likely to contain cancer, and to sample from these during biopsy. The test that is currently gaining most favour in conferring this information is multiparametric MRI (mpMRI).[9]

An MRI-based approach to diagnosis would require all men with raised PSA to have an mpMRI. Men who are negative on mpMRI would receive no further biopsy. The men with a suspicious lesion on mpMRI would undergo an MRI-targeted biopsy. During MRI-targeted biopsy, the biopsy needle can be directed by the clinician using prior mpMR images and real-time ultrasound ('visual' or 'cognitive registration'), by using assistive technology that digitally overlays the target information derived from the mpMRI directly onto the ultrasound image ('computer-aided registration' or 'image fusion') or the biopsy can be performed within the MR scanner itself ('in-bore biopsy' or 'MR-guided MR biopsy'). Irrespective of the image-guided technique, an mpMRI-based approach to diagnosis has three potential advantages. First, patients with no lesion on mpMRI could avoid a prostate biopsy. Second, patients with clinically insignificant disease would avoid diagnosis and subsequent inappropriate treatment which carries risk of side effects and no benefit in terms of survival. Third, using mpMRI for targeting may improve the detection of clinically significant cancers and improve risk stratification.

The UK's National Institute for Health and Care Excellence (NICE) has recently acknowledged the utility of mpMRI, but stopped short of a recommendation to offer prebiopsy mpMRI to all men.[7] It remains controversial partly due to doubts about the performance and reproducibility of mpMRI. Despite this, many providers have adopted an image-guided biopsy approach in response to a man presenting with an elevated PSA.[10]

We summarise what can be understood from recently emerging evidence in a format that is relevant for clinical decision making. We carried out a decision analysis to compare a simplified version of the current standard diagnostic approach (TRUS-guided biopsy) with an approach where mpMRI is used to select men for biopsy and to guide the biopsy needle towards the area of suspected cancer. We estimate the number of biopsies that could be avoided with prebiopsy mpMRI and the number of correctly identified patients with and without clinically significant prostate cancer.

## METHODS
### Decision analysis
We used a decision tree to compare the standard diagnostic pathway (TRUS-guided biopsy for all) with a new pathway (mpMRI for all, then MRI-targeted biopsy if positive). The tree, presented in figure 1, was evaluated to reveal the expected outcomes associated with each option, for a hypothetical cohort of 1000 men by multiplying the prevalence estimates of our target condition by sensitivity and specificity estimates of the diagnostic tests. The test accuracy estimates used to populate the decision tree were derived from recent studies which reported data that reflected the conditional nature of the parameters and used an appropriate reference test.[11] [12] All of these data are limited in some way, but assumptions were made so that any biases would favour the current diagnostic approach.

### Target population
The target population for the decision analysis was men with increased serum PSA levels or abnormal findings on digital rectal examination who had never had a prostate biopsy.

### Clinically significant disease
For our base-case analysis we defined clinically significant disease according to widely used, and arguably somewhat conservative, criteria: a minimum volume of 0.2 cc or cell differentiation corresponding to a Gleason

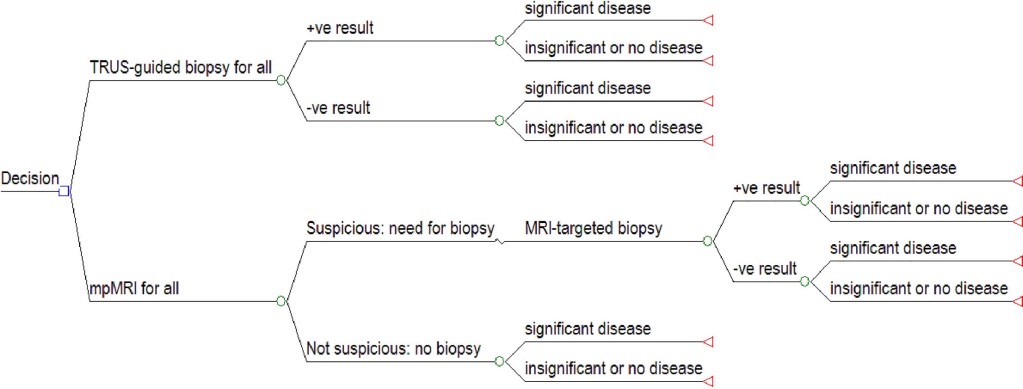

**Figure 1** Structure of the decision tree.

score of 3+4 or higher.[13] The prevalence of clinically significant disease in our target population is uncertain, but we estimated it to be 50% of all men with suspected prostate cancer, based on a prospective analysis of men undergoing a first prostate biopsy.[14] The remaining 50% are assumed to have clinically insignificant disease or no cancer. We varied the prevalence of clinically significant disease in a sensitivity analysis.

## TRUS-guided biopsy

The gold standard used to establish the presence or absence of clinically significant disease—whole-mount pathological data—is usually only available for men who test positive and then go on to have radical surgery.[7 11 15] Therefore, we used data from a study which carried out computer simulations to estimate the performance characteristics of TRUS-guided biopsy by comparing them to reconstructed whole-mount pathology obtained from patients undergoing surgery for bladder cancer, which revealed that they also had prostate cancer.[16] The spectrum of disease in this sample population is likely to include more early-stage disease than would be expected in an unscreened UK population, and thus this bias will favour the current diagnostic approach.

The sensitivity of TRUS-guided biopsy, when criteria proposed by Epstein were used to interpret the diagnostic result, was approximately 50%.[16] According to Epstein et al,[13] a biopsy result is positive for significant cancer if the maximum cancer core length from biopsy is at least 3 mm or if the Gleason score is 3+4 or higher. The corresponding specificity of TRUS-guided biopsy was estimated to be approximately 90%, which represents the proportion of men correctly identified with insignificant disease (men with no prostate cancer were not included in the Lecornet et al[16] study population).

## mpMRI and MRI-targeted biopsy

We estimated the diagnostic accuracy for the MRI-based strategy by combining the test accuracy estimates for mpMRI and MRI-targeted biopsy. A recent systematic review of the literature revealed two studies on mpMRI in biopsy-naïve men with suspected prostate cancer.[11 17 18] Only one of these studies reported data at the level of detail required to estimate sensitivity and specificity of mpMRI: a large study involving 555 men which compared prebiopsy mpMRI results with TRUS-guided biopsy and/or MRI-targeted biopsy as a proxy for true disease status.[17] We used these data to estimate the sensitivity of mpMRI at 80% and the specificity at 60%. A more recent study shows these values may in fact underestimate the performance of mpMRI.[14]

The accuracy of MRI-targeted biopsy was taken from a study that compared MRI-targeted biopsy with cognitive registration to 20-sector template-prostate mapping.[19] This study was used since all men in the study population had a lesion on mpMRI and therefore allowed us to capture the sequential nature of the diagnostic approach.

The study showed that when the biopsies were classified according to the Epstein criteria, the sensitivity was approximately 80% and the specificity 80%.[19] The specificity that this study reported for the MRI-targeted biopsy is lower than our estimate of the specificity of TRUS-guided biopsy (90%).[19] However, the use of MRI-targeting instead of TRUS-guided biopsy should have no impact on men without clinically significant disease, and therefore we assumed that MRI-targeted biopsy should be as good as—but not better than—TRUS-guided biopsy at correctly identifying men without clinically significant prostate cancer. We therefore used 90% as the specificity estimate for MRI-targeted biopsy in the decision analysis, the same as that of TRUS-guided biopsy. We assessed the impact this had on the overall diagnostic results in a sensitivity analysis.

## Sensitivity analysis

We carried out a one-way sensitivity analysis by varying the prevalence of clinically significant disease from 0 to 1, keeping all other variables constant. This sensitivity analysis was intended to demonstrate the extent to which the optimal diagnostic strategy depends on the prevalence of clinically significant disease. We also carried out two sensitivity analyses to assess the impact of specific test performance estimates of the mpMRI-based strategy on overall diagnostic outcomes. In the first scenario (scenario i) we used a pooled estimate of mpMRI test performance from a recent meta-analysis (mpMRI sensitivity 74%, mpMRI specificity 88%).[20] In the second scenario (scenario ii) we investigated the impact of our assumption that MRI targeting has no impact on men without clinically significant disease (by using a specificity of 80% for MRI-targeted biopsy as reported by Kasivisvanathan et al[19] rather than our base case estimate of 90%).

Although these sensitivity analyses provide some insight into the specific impact of individual parameters, the estimates used to describe the performance of all the diagnostic tests are associated with significant uncertainties. Therefore, to assess the robustness of our results, we performed a probabilistic sensitivity analysis using Monte Carlo simulation by varying the sensitivities and specificities of the three tests (TRUS-guided biopsy, mpMRI and MRI-targeted biopsy) simultaneously over 2000 iterations, sampling from beta distributions to characterise the uncertainty in the test accuracy data (see table 1). We determined the beta distributions by assuming that the sensitivities and specificities were observed in populations consisting of 50 men with and 50 men without clinically significant disease. We substantially widened the distributions (by assuming a small population of men) in order to increase the uncertainty associated with the test performance parameters. We ignored the correlation between sensitivity and specificity and kept the disease prevalence constant at 50%.

Table 1 Diagnostic accuracy estimates of TRUS-guided biopsy, mpMRI and MRI-targeted biopsy used in the base case analysis

| Index test | Sensitivity | Specificity | Reference test | Source and patient population |
|---|---|---|---|---|
| TRUS-guided biopsy | **50%** (16/34 patients, 95% CIs 27% to 73%) $\alpha=25, \beta=25$ | **90%** (57/62 patients, 95% CIs 78% to 97%) $\alpha=45, \beta=5$ | Whole-mount pathology | Lecornet et al[16]: simulated biopsy results on digitally reconstructed prostates of 96 men who had undergone surgery for bladder cancer which revealed prostate cancer |
| mpMRI | **80%** (252/302, 95% CIs 66% to 90%) $\alpha=40, \beta=5$ | **60%** (154/253 patients, 95% CIs 45% to 76%) $\alpha=30, \beta=20$ | TRUS-guided extended systematic biopsies (10–12 core) plus two targeted biopsies for those with any area suspicious on mpMRI (score ≥3) | Haffner et al[17]: 555 men with suspected localised prostate defined as raised PSA of >3–4 ng/mL and/or abnormal DRE with no clinical or biological suspicion of stage T>3 or metastases and had no prior biopsy |
| MRI-targeted biopsy | **80%** (94/121 patients, 95% CIs 66% to 90%) $\alpha=40, \beta=10$ | **90%** Assumed to be equivalent to the specificity of TRUS-guided biopsy, (57/62 patients, 95% CIs 78% to 97%) $\alpha=45, \beta=5$ | 20 sector-TPM | Kasivisvanathan et al[19]: 182 men who had a suspicious lesion on mpMRI; 78 of whom were biopsy naive, 32 had a prior negative biopsy and 72 had a prior positive biopsy |

TRUS, transrectal ultrasound; TPM, template mapping biopsy; mpMRI, multiparametric MRI; MRI-TB, MRI-targeted biopsy. Data inputs were rounded to the nearest 5%. Beta distributions were estimated using the integer form in Excel according to the parameters α and β.

## RESULTS

The decision tree estimated that the use of TRUS-guided biopsy in a hypothetical cohort of 1000 men with suspected prostate cancer—and an estimated 50% prevalence of clinically significant disease—would result in 300 positive and 700 negative biopsy results, which would correctly identify 250 men with clinically significant prostate cancer and 450 men without the disease (table 2). It follows that 250 men with significant prostate cancer would be missed by TRUS-guided biopsy and 50 men who do not have significant prostate cancer would wrongly receive a diagnosis.

The use of mpMRI and MRI-targeted biopsy in the same cohort would result in 600 men undergoing a prostate biopsy with 340 positive and 260 negative biopsy results (table 2). This strategy would correctly identify 320 men as having significant prostate cancer and 480 without the disease. In other words, the use of the mpMRI-based strategy would fail to diagnose significant cancer in 180 men (500–320), which is the result of significant prostate cancers that were missed by mpMRI in addition to significant cancers that were identified on mpMRI but were missed by MRI-targeted biopsy. Twenty men (500–480) who do not have clinically significant prostate cancer would wrongly receive a diagnosis.

mpMRI followed by MRI-targeted biopsy can be said to 'clinically dominate' TRUS-guided biopsy as the strategy results in fewer expected biopsies (600 vs 1000), more men being correctly identified as having clinically significant disease (320 vs 250), and fewer men being falsely identified with the disease (20 vs 50).

Figure 2 provides a visual representation of the one-way sensitivity analysis showing the total number of people receiving the wrong diagnosis (the sum of the number of patients with a false-positive or a false-negative result) as a function of the prevalence of clinically significant disease. The mpMRI-based approach resulted in a lower number of patients wrongly diagnosed than with TRUS-guided biopsy for all men, at all prevalence rates. Below a prevalence of 5%, doing nothing is the 'optimal' strategy as it leads to the lowest number of men with the wrong diagnosis. Above a prevalence of 70%, treating all men is optimal.

Table 3 demonstrates that assuming a sensitivity of 74% and a specificity of 88% as estimates of the test performance of mpMRI (instead of a sensitivity of 80% and a specificity of 60% used in the base case analysis) found that the mpMRI-based strategy resulted in far fewer biopsies than our base case estimation (430 instead of 600), but slightly worse diagnostic outcomes for men with significant disease (296 correctly identified instead of 320), which was still better than the current standard of care (250 correctly identified). Table 3 also shows that using a lower specificity for MRI-targeted biopsy (80% instead of 90%) resulted in 40 men (instead of 20) without clinically significant disease wrongly identified as having significant cancer, but this is still less than the 50 men who would be wrongly identified with the standard diagnostic approach using TRUS-guided biopsy alone.

**Table 2** Details of calculations and results of the decision analysis for a cohort of 1000 men comparing TRUS-guided biopsy with mpMRI and MRI-targeted biopsy

| | TRUS-guided biopsy for all | mpMRI then MRI-targeted biopsy |
|---|---|---|
| Number of biopsies | **1000** <br><br> (all men) | **600** <br> $= P(MRI^+\|D^+)+P(MRI^+\|D^-)$ <br> $= (MRI_{sens}*prev*no\_in\_cohort)+$ <br> $((1-MRI_{spec})*(1-prev)*no\_in\_cohort))$ <br> $= (0.8*0.5*1000)+((1-0.6)*(1-0.5)*1000)$ |
| Patients with clinically significant cancer and correctly identified (true positive) | **250** <br> $=P(TRUS^+\|D^+)$ <br> $=TRUS_{sens}*prev*no\_in\_cohort$ <br> $=0.5*0.5*1000$ | **320** <br> $=P(MRI^+\|D^+).P(MRITB^+\|D^+)$ <br> $=MRI_{sens}*MRITB_{sens}*prev*no\_in\_cohort$ <br> $=0.8*0.8*0.5*1000$ |
| Patients with clinically significant cancer and wrongly identified (false negative) | **250** <br> $=p(TRUS^-\|D^+)$ <br> $=(1-TRUS_{sens})*prev*no\_in\_cohort$ <br> $=(1-0.5)*0.5*1000$ | **180** <br> $=P(MRI^-\|D^+)+P(MRI^+\|D^+).P(MRITB^-\|D^+)$ <br> $=((1-MRI_{sens})*prev*no\_in\_cohort) +$ <br> $(MRI_{sens}*(1-MRITB_{sens})*prev*no\_in\_cohort)$ <br> $=((1-0.8)*0.5*1000)+(0.8*(1-0.8)*0.5*1000)$ |
| Patients with insignificant prostate cancer or no prostate cancer and correctly identified (true negative) | **450** <br> $=P(TRUS^-\|D^-)$ <br> $=TRUS_{spec}*(1-prev)*no\_in\_cohort$ <br> $=0.9*(1-0.5)*1000$ | **480** <br> $=P(MRI^-\|D^-)+P(MRI^+\|D^-).P(MRITB^-\|D^-)$ <br> $=(MRI_{spec}*(1-prev)*no\_in\_cohort) +$ <br> $((1-MRI_{spec})*MRITB_{spec}*(1-prev)*no\_in\_cohort)$ <br> $=(0.6*(1-0.5)*1000)+((1-0.6)*0.9*(1-0.5)*1000)$ |
| Patients with insignificant prostate cancer or no prostate cancer and wrongly identified (false positive) | **50** <br> $=P(TRUS^+\|D^-)$ <br> $=(1-TRUS_{spec})*(1-prev)*no\_in\_cohort$ <br> $=(1-0.9)*0.5*1000$ | **20** <br> $=P(MRI^+\|D^-).P(MRITB^+\|D^-)$ <br> $=(1-MRI_{spec})*(1-MRITB_{spec})*(1-prev)*no\_in\_cohort$ <br> $=(1-0.6)*(1-0.9)*(1-0.5)*1000$ |

'prev' – prevalence; 'no_in_cohort' – number of men in cohort; 'TRUS$_{sens}$' – sensitivity of TRUS-guided biopsy; 'TRUS$_{spec}$' – specificity of TRUS-guided biopsy; 'MRI$_{sens}$' – sensitivity of mpMRI; 'MRI$_{spec}$' – specificity of mpMRI; 'MRITB$_{sens}$' – sensitivity of MRI-targeted biopsy; 'MRITB$_{spec}$' – specificity of MRI-targeted biopsy.

When the sensitivities and specificities of the three tests were varied simultaneously in 2000 simulations for the probabilistic sensitivity analysis, the diagnostic approach using mpMRI and MRI-targeted biopsy clinically dominated 86% of the simulations, whereas TRUS-guided biopsy dominated in 0.8% of the simulations. Within the remaining 13.2% of simulations, the choice between the mpMRI-based strategy and TRUS-guided biopsy is not clear as there was a 'trade-off' between outcomes. That is, either the

mpMRI-based strategy correctly identified more men with clinically significant cancer but fewer men without clinically significant disease than TRUS-guided biopsy, or vice versa.

## DISCUSSION
Our decision analysis suggests that mpMRI of the prostate followed by MRI-targeted biopsy if positive could result in fewer and better biopsies than a strategy using

**Figure 2** One-way sensitivity analysis showing the expected number of patients with wrong diagnoses according to the prevalence of clinically significant disease in a cohort of 1000 men. See text for further explanation.

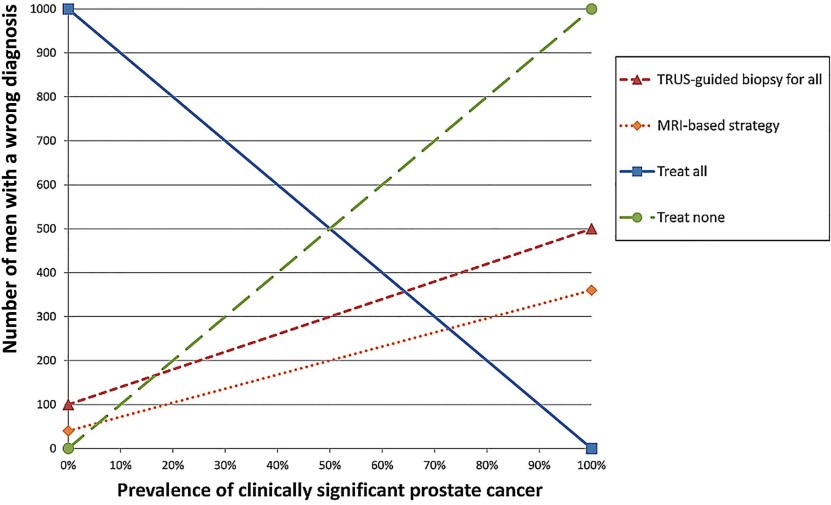

**Table 3** Results of sensitivity analyses in a cohort of 1000 men

| Scenario | Base case analysis | | Scenario i (mpMRI sensitivity 74%, specificity 88%) | | Scenario ii (MRI-targeted biopsy sensitivity 80%, specificity 80%) | |
|---|---|---|---|---|---|---|
| Strategy | TRUS-guided biopsy for all | mpMRI then MRI-targeted biopsy | TRUS-guided biopsy for all | mpMRI then MRI-targeted biopsy | TRUS-guided biopsy for all | mpMRI then MRI-targeted biopsy |
| Number of biopsies | 1000 | 600 | 1000 | 430 | 1000 | 600 |
| Patients with clinically significant cancer and correctly identified (true positive) | 250 | 320 | 250 | 296 | 250 | 320 |
| Patients with clinically significant cancer and wrongly identified (false negative) | 250 | 180 | 250 | 204 | 250 | 180 |
| Patients with insignificant prostate cancer or no prostate cancer and correctly identified (true negative) | 450 | 480 | 450 | 494 | 450 | 460 |
| Patients with insignificant prostate cancer or no prostate cancer and wrongly identified (false positive) | 50 | 20 | 50 | 6 | 50 | 40 |

mpMRI, multiparametric MRI; TRUS, transrectal ultrasound.

only TRUS-guided biopsy. The results suggest that the mpMRI-based strategy could reduce the number of biopsies by about one-third (600 compared to 1000 biopsies), increase the number of men identified with clinically significant cancer by about 30% (320 compared to 250 patients), and reduce the number of men falsely identified with the disease by 60% (20 compared to 50). These results are in line with those of recent clinical studies comparing similar strategies, albeit not against a gold standard of pathology or template biopsy.[21][22]

When we accounted for uncertainty in the sensitivity and specificity estimates of the three diagnostic tests, we found that the dominance of the mpMRI-based strategy was robust. However, the probabilistic sensitivity analysis did not assess the impact of inherent 'structural' uncertainties, such as the ongoing debate about the definition of clinically significant cancer, various diagnostic thresholds used to decide whether mpMRI or biopsy results are positive or negative, and the use of imperfect gold-standard tests.

NICE has recently updated its guidance on the diagnosis and management of men with prostate cancer.[7] Our estimate of the sensitivity of 50% for TRUS-guided biopsy is close to the estimate of 45% used in its analysis.[7] However, we estimated the specificity of the TRUS-guided biopsy to be 90%, while NICE assumed it to be 100%. While both of these specificity estimates are somewhat speculative, we believed that the specificity estimate needed to reflect that patients who have clinically insignificant prostate cancer may have biopsy results that are interpreted as being suggestive of clinically

significant cancer. We also assumed that this 'error' is as likely with TRUS-guided as with MRI-targeted biopsies and therefore we used the same false-positive rate for TRUS-guided and for MRI-targeted biopsies. In addition we used data on the test accuracy of MRI-targeted biopsy from a study which used visual registration techniques. It has been suggested that computer-aided registration techniques may be more accurate,[7] although recent randomised controlled trial (RCT) data showed no statistically significant difference in detection rates.[23]

The results of our analysis are based on a simplification of the choices facing urologists in the diagnosis of prostate cancer. In their evaluation, NICE considered a strategy of mpMRI and biopsy for all men, including targeted biopsies for all men with a lesion on mpMRI. This perhaps highlights the reticence of healthcare professionals to do 'less' rather than 'more', which may be influenced by concern over medical liability. A major challenge therefore will be the implementation of a strategy that requires a negative diagnostic test result to be established and then followed by no immediate further investigation. New guidelines based on the results of forthcoming RCTs (such as PROMIS) and expert consensus may be required to avoid a 'creep' in the numbers of unnecessary biopsies.[24]

In this analysis we focused purely on short-term clinical outcomes following different testing options. Ultimately, however, the optimal diagnostic strategy for men with suspected prostate cancer will depend on the impact on costs and quality-adjusted life expectancy. The cost of the diagnostic procedures may in fact be about

the same for the two diagnostic strategies. If all men receive an mpMRI (£200 in 2011–2012 UK National Health Service (NHS) prices) and 60% of these men also receive a biopsy (£540 in 2011–2012 UK NHS prices) the mpMRI-based strategy will result in an average cost of £524 per man, assuming that a TRUS-guided biopsy and MRI-targeted biopsy are equivalent in cost.[25] This compares to £540 per man with a TRUS-guided biopsy. Of course, the true costs of the two strategies include the long-term costs and consequences of further investigations and treatments which need to be taken into account in future economic modelling. Initial estimates from a published economic evaluation suggest that an mpMRI-based strategy is likely to be highly cost effective in the Netherlands,[26] although uncertainties, particularly around long-term health outcomes, remain.[27]

Despite the complexity of the downstream pathways, estimates of diagnostic performance and disease prevalence will be key drivers of the clinical and cost effectiveness of the whole of prostate cancer care. Systematic reviews of the prostate biopsy and imaging literature have revealed a large number of small studies characterised by poor reporting and important biases.[7 11 12 15] In our analysis we only used very recently published studies that capture the emerging evidence on how well TRUS-guided biopsy, mpMRI and MRI-targeted biopsy perform and estimate disease prevalence. Future research efforts in prostate cancer need to focus on providing accurate and precise estimates of these parameters. Studies need to consistently distinguish between significant and insignificant cancer, represent the sequential nature of diagnostic tests and should adhere to high standards of reporting such as the START guidelines for MRI.[28 29] Without these studies, it will be hard to accurately evaluate the role of targeted biopsy or any new strategy for diagnosing prostate cancer in future.

## CONCLUSIONS
Our analysis suggests that mpMRI followed by MRI-targeted biopsy may result in fewer and better biopsies than TRUS-guided biopsy for all men. We found that the mpMRI-based strategy correctly identified more men with significant prostate cancer and also correctly identify more men without the disease in 86% of the simulations in our probabilistic sensitivity analysis. Estimates of disease prevalence and diagnostic performance will be key drivers of a full economic analysis, so research efforts should focus on providing precise estimates of these crucial parameters.

**Acknowledgements** The authors would like to thank Yipeng Hu and Veeru Kasivisvanathan for their assistance in providing additional data for the analysis. They also acknowledge the research support Mark Emberton receives from the United Kingdom's National Institute of Health Research UCL/UCLH Biomedical Research Centre, London. This publication presents independent research commissioned by the Health Innovation Challenge Fund (HICF-T4–310), a parallel funding partnership between the Wellcome Trust and the Department of Health.

**Contributors** JvdM, SRW, AHM, HUA and ME made substantial contributions to the conception and design of the work; SRW, HUA, CMM and ID acquired the data; SRW, JvdM, HUA, CMM and ID analysed and interpreted the data; SRW and JvdM drafted the work and HUA, CMM, ID, ME and AHM provided critical revision of the manuscript; ME, HUA, CMM, JvdM, SRW and AHM obtained funding and JvdM and AHM provided supervision. All authors have read and approved the final manuscript and agree to be accountable for all aspects of the work.

**Funding** This work was funded by the Health Innovation Challenge Fund (Wellcome Trust and UK Department of Health) Award Number: HICF-T4-310.

**Competing interests** All authors have completed the ICMJE uniform disclosure at http://www.icmje.org/coi_disclosure.pdf and declare: financial support for the submitted work from the Wellcome Trust and the Department of Health (for all authors); ME is a consultant/investigator for USHIFU, STEBA Biotech, Sanofi Aventis, GlaxoSmithKline and Angiodynamics, he is also a consultant for Sophiris and is a director of Mediwatch PLC. HUA is an investigator for USHIFU, Angiodynamics and Advanced Medical Diagnostics (AMD). CMM has received research grants from GlaxoSmithKline, the Wellcome Trust and Advanced Medical Diagnostics (AMD), is a consultant/investigator for STEBA Buiotech and has been paid for lecturing by Sanofi. JvdM, AHM and SRW have no other interests to disclose.

**Ethics approval** None required.

**Provenance and peer review** Not commissioned; externally peer reviewed.

**Data sharing statement** The data inputs, the decision tree structure and the calculations are given in full so no additional data are available.

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
