## [Reviewer comments · BMJ Open]

Some articles will have been accepted based in part or entirely on reviews undertaken for other BMJ Group journals. These will be reproduced where possible.

ARTICLE DETAILS

TITLE (PROVISIONAL)	Multi-parametric MRI followed by targeted prostate biopsy for men with suspected prostate cancer: a clinical decision analysis
AUTHORS	Willis, Sarah; Ahmed, Hashim; Moore, Caroline; Donaldson, Ian; Emberton, Mark; Miners, Alec; van der Meulen, Jan

VERSION 1 - REVIEW

REVIEWER	Timur Kuru University of Heidelberg
REVIEW RETURNED	07-Mar-2014

GENERAL COMMENTS	The authors present a well written paper clinical decision analysis on mp-MRI followed by targeted biopsy and compare this to standard TRUS-biopsy. The cost estimations are calculated for GB health system and are very promising for the widespread of MRI/TRUS fusion biopsy. As it is a hypothetical cohort study, it would benefit from statistical review. I am personally not confident enough to judge the statistical analyses. There are some mistakes which should be corrected: 1. The limitation section states, that there is no study comparing TRUS and fusion biopsy. There is actually one study doing that: Siddiqui MM, et al. Magnetic resonance imaging/ultrasound-fusion biopsy significantly upgrades prostate cancer versus systematic 12-core transrectal ultrasound biopsy. European Urology. 2013 Nov;64(5):713–9. This study should be included and discussed. The limitation section should be changed accordingly.2. Reference 11 and 16 are the same. Reference 6 is outdated and should be corrected to the most recent guideline.
---

REVIEWER	Maarten de Rooij Radboud university medical center, Nijmegen, The Netherlands
REVIEW RETURNED	11-Mar-2014

GENERAL COMMENTS	Willis and colleagues use a decision analysis to compare the current diagnostic pathway with an MRI based pathway for the diagnosis of prostate cancer. The authors should consider the following points. 1. The assumption that 50% have clinically significant disease, while the other 50% have non-significant disease in a group of 1000 men with an elevated PSA should be explained in more detail. What
--

	about men without prostate cancer? Are these men part of the 50% defined as "non-significant disease"? 2. One of the major advantages of mpMRI followed by MRI targeted biopsy is not only the avoidance of unnecessary biopsy and the improvement of detection of clinically significant disease. The potential to reduce diagnosis of insignificant disease, which could improve over-diagnosis and over-treatment, is also a major advantage of an image based strategy. The authors should address this in the introduction/discussion of the manuscript. 3. MRI targeted biopsy can be performed in 3 ways: 1) "cognitive registration" 2) MRI/TRUSGB fusion 3) MR guided MR biopsy. This last group is missing in the introduction of the manuscript. 4. Recently a cost-effectiveness analysis of our group comparing the current diagnostic strategy with an MRI based strategy is published in Eur Urol. The authors should consider referring this paper in the introduction or discussion of the manuscript. 5. Recently our group published a diagnostic meta-analysis of the detection of Pca by mpMRI in AJR. We found a pooled specificity of 88% and sensitivity of 74%. The authors should consider including this study in the evidence syntheses of the assumption on diagnostic accuracy of mpMRI. 6. The authors should be aware of a study that is currently in press at Eur Urol. In this prospective diagnostic study the current diagnostic strategy is compared with an MRI based strategy in a biopsy-naive cohort of men with an elevated PSA/suspicious DRE. The results of this study are in line with the results of the decision analysis presented in this manuscript. 7. "Cancer grade" should be rephrased as "Gleason score" (introduction). Although the reviewer is not asked to give an opinion on priority or breadth of appeal, I would like to mention that the results of this study were already published in several newspapers/online media (dailymail, the telegraph, etc) in June 2013.
--	---

REVIEWER	Peter Choyek NCI, USA
REVIEW RETURNED	14-Mar-2014

GENERAL COMMENTS	Summary: A decision analysis is conducted between TRUS biopsy and MR guided biopsy in 1000 theoretical patients. MR with guided biopsy is shown to clinically dominate in the majority of scenarios. Comment: Studies of this type are valuable in setting the groundwork for actual efficacy studies. If a reasoned model fails to show benefit of a new technology, there is less impetus to invest the huge resources needed to study it in real patients. Thus, this is an important contribution to the literature. 1. One problem is the definition of an actionable MRI is still debated and probably varies according to the health care system. In the US, for instance, there will be a bias toward biopsy because of a concern over medical liability. This could result in more MRI biopsies than the European literature might predict. Please comment
--

	2. Define the actionable elevated PSA level in these studies 3. The input data for the MR biopsy seems to rely on a study that used cognitive fusion. This is highly user specific and probably not as reliable as a software solution, rendering the MRI and biopsy less reliable. The model should incorporate computer aided fusion. This should reduce the number of false negatives. Please add as a weakness to the study. 4. Please explain the 18% false negative rate for MRI (significant tumors) in Table 2. This is higher than I would have predicted. Is this a result of pathology error? Please explain in more depth how there would be 180 false negatives for significant disease on MRI. 5. A problem with basing the assumptions on different published studies is that each study has different disease prevalence rates. For instance, referral hospitals (which often do most of the publishing) may have higher rates (or lower rates) of clinically significant disease compared with a community hospital. This depends highly on how a patient ends up in a certain hospital. It would be preferable to use a collection of single institution data (for TRUS and MRI). The merits of both approaches should be discussed. 6. I did not follow the reasoning behind changing the MRI specificity from the reported level of 60% to 90%. I agree that the 60% specificity number is incorrect but it seems a bit ad hoc to simply change it to 90%. What happens if you run it at 60%? What about referencing other studies (especially since this was based on cognitive fusions) 7. To avoid confusion on page 11, all of the 4 outcomes should be listed (referenced to Table 2). Otherwise there is a question about 300 patients not mentioned. 8. Somewhere the term "clinically dominate" should be defined. 9. Wonder if it would be possible to create an online "calculator" in which other investigator's data for TRUS and MR guided bx could be entered. If you kept this centrally you could generate world wide, a lot of data that could generalize across many health care systems and would eliminate patient selection biases by averaging. The input data from the literature is highly selective and could influence the output. The Excel One program might be a simple way to do this.
--	--

REVIEWER	Andrew Martin NHMRC Clinical Trials Centre, University of Sydney
REVIEW RETURNED	16-Apr-2014

GENERAL COMMENTS	Page 3 line 48: Replace "...demonstrates..." with "...suggests..." Page 10 line 36-41: Report the shape parameters of the beta distributions, or indicate the 2.5% and 97.5% percentile from the distributions specified. Page 11 line 7: Replace "...demonstrated..." with "...estimated..." Page 12 line 17-22: provide details on the results of the 9.6% of simulations where trade-offs arose. Page 13 line 8: Replace "...revealed..." with "...suggests..." Page 13 line 12: Replace "...would..." with "...could possibly..." Page 14 line 13: The specificity estimate for MRI targeted biopsy was 'bumped up' to match the 90% for TRUS biopsy (page 10 line 3). This would improve the comparative effectiveness of mpMRI rather than underestimate it (as suggested on page 14 line 13). Page 16 line 5: Replace "...demonstrates..." with "...suggests..." and
---

VERSION 1 – AUTHOR RESPONSE

Reviewer 1: Timur Kuru, University of Heidelberg

The authors present a well written paper clinical decision analysis on mp-MRI followed by targeted biopsy and compare this to standard TRUS-biopsy.

The cost estimations are calculated for GB health system and are very promising for the widespread of MRI/TRUS fusion biopsy.

As it is a hypothetical cohort study, it would benefit from statistical review. I am personally not confident enough to judge the statistical analyses.

There are some mistakes which should be corrected:

1. The limitation section states, that there is no study comparing TRUS and fusion biopsy. There is actually one study doing that: Siddiqui MM, et al. Magnetic resonance imaging/ultrasound-fusion biopsy significantly upgrades prostate cancer versus systematic 12-core transrectal ultrasound biopsy. *European Urology*. 2013 Nov;64(5):713–9.

This study should be included and discussed. The limitation section should be changed accordingly. Siddiqui 2013 paper compares TRUS with mpMRI-targeted biopsy but not against a gold standard (e.g. whole mount pathology) and thus sensitivity and specificity estimates were not calculated. We have amended the limitations section to say that a study comparing these techniques against a gold standard has not been conducted. (Strengths and limitations section, page 5, line 76). We also refer to the paper in the Discussion, page 17, lines 300-301. Reference 22.

2. Reference 11 and 16 are the same. Reference 6 is outdated and should be corrected to the most recent guideline.

Duplicate reference removed. Reference 6 has been corrected to the 2013 update of the EAU guidelines (published 2014). We have also changed reference 7 (NICE guideline) which has been updated since date of submission.

Reviewer 2: Maarten de Rooij, Radboud university medical center, Nijmegen, The Netherlands

Willis and colleagues use a decision analysis to compare the current diagnostic pathway with an MRI based pathway for the diagnosis of prostate cancer. The authors should consider the following points.

1. The assumption that 50% have clinically significant disease, while the other 50% have non-significant disease in a group of 1000 men with an elevated PSA should be explained in more detail. What about men without prostate cancer? Are these men part of the 50% defined as "non-significant disease"?

Of the initial population with elevated PSA, we assume 50% have clinically significant prostate cancer and the remaining 50% will have either clinically insignificant prostate cancer or no prostate cancer at all. We have added a sentence to clarify this in the Methods section on page 10, paragraph 1, lines 161-162. "Non-significant disease" is misleading as a term, since this includes men with no cancer. We have removed this term from figure 1 and table 2. Elsewhere the text refers to men with clinically significant prostate cancer or without clinically significant prostate cancer.

2. One of the major advantages of mpMRI followed by MRI targeted biopsy is not only the avoidance of unnecessary biopsy and the improvement of detection of clinically significant disease. The potential to reduce diagnosis of insignificant disease, which could improve over-diagnosis and over-treatment, is also a major advantage of an image based strategy. The authors should address this in the introduction/discussion of the manuscript.

In fact the strategy we propose would minimise or eliminate the diagnosis of clinically insignificant disease. We have added this as a potential advantage of the MRI-approach, in the Introduction (Page 7, paragraph 1, lines 119-121).

3. MRI targeted biopsy can be performed in 3 ways: 1) "cognitive registration" 2) MRI/TRUSGB fusion 3) MR guided MR biopsy. This last group is missing in the introduction of the manuscript. We have added this technique, also known as in-bore biopsy to the Introduction (on page 7, paragraph 1, lines 115-116).

4. Recently a cost-effectiveness analysis of our group comparing the current diagnostic strategy with an MRI based strategy is published in Eur Urol. The authors should consider referring this paper in the introduction or discussion of the manuscript. We have added this to the Discussion where we discuss potential for cost-effectiveness (Page 19, paragraph 1, lines 346-348. Reference 27).

5. Recently our group published a diagnostic meta-analysis of the detection of Pca by mpMRI in AJR. We found a pooled specificity of 88% and sensitivity of 74%. The authors should consider including this study in the evidence syntheses of the assumption on diagnostic accuracy of mpMRI. We have conducted a sensitivity analysis to see the impact on our results of using sensitivity and specificity of mpMRI from this meta-analysis (labelled 'scenario i'). As expected the results slightly differed from our base case but benefit of the MRI-based strategy over TRUS-guided biopsy only remained. The methods are presented on page 12, paragraph 2, lines 215-219 and the results are presented on page 15, paragraph 2, lines 272-277. The results of this scenario are also presented in table 3.

6. The authors should be aware of a study that is currently in press at Eur Urol. In this prospective diagnostic study the current diagnostic strategy is compared with an MRI based strategy in a biopsy-naive cohort of men with an elevated PSA/suspicious DRE. The results of this study are in line with the results of the decision analysis presented in this manuscript. This study has been included in the Discussion (page 17, paragraph 1, lines 300-301. Reference 21).

7. "Cancer grade" should be rephrased as "Gleason score" (introduction). Done (Introduction, page 6, paragraph 1, line 98).

Although the reviewer is not asked to give an opinion on priority or breadth of appeal, I would like to mention that the results of this study were already published in several newspapers/online media (Dailymail, the telegraph, etc) in June 2013.

The newspaper article was based on a poster presentation of work in progress at the (UK) National Cancer Intelligence Network Cancer Outcomes conference. A journalist quoted a number of preliminary findings out of context. The reviewer's claim that the results have already been published are unfounded.

Reviewer 3: Peter Choyek NCI, USA

Summary: A decision analysis is conducted between TRUS biopsy and MR guided biopsy in 1000 theoretical patients. MR with guided biopsy is shown to clinically dominate in the majority of scenarios.

Comment: Studies of this type are valuable in setting the groundwork for actual efficacy studies. If a reasoned model fails to show benefit of a new technology, there is less impetus to invest the huge resources needed to study it in real patients. Thus, this is an important contribution to the literature.

1. One problem is the definition of an actionable MRI is still debated and probably varies according to the health care system. In the US, for instance, there will be a bias toward biopsy because of a concern over medical liability. This could result in more MRI biopsies than the European literature might predict. Please comment

Added to the Discussion (page 18, paragraph 2, line 330 and lines 332-334) where we mention concerns with regards liability as a factor that may have an effect on decision to perform biopsies.

2. Define the actionable elevated PSA level in these studies

Actionable elevated PSA level for the mpMRI-based strategy, based on Haffner 2011 paper has been added to table 1 (patients in this study had raised PSA defined as >3-4ng/ml and/or abnormal DRE). Actionable elevated PSA is not relevant for patients from the other studies used to populate the decision tree (Kasivisvanathan et al 2013 and Lecornet et al 2012).

3. The input data for the MR biopsy seems to rely on a study that used cognitive fusion. This is highly user specific and probably not as reliable as a software solution, rendering the MRI and biopsy less reliable. The model should incorporate computer aided fusion. This should reduce the number of false negatives. Please add as a weakness to the study.

There is evidence (summarised in the recent NICE guideline) to suggest image-fusion biopsies are more accurate than those performed using cognitive registration techniques, although recent RCT evidence (Wysock et al) shows no statistically significant difference in cancer detection rates between targeted biopsy using visual registration and image-fusion techniques. If image-fusion biopsies are more accurate, then the estimates we use will have underestimated the potential benefit of a strategy using MRI-targeted biopsy. We have added this to the Discussion section (page 18, paragraph 1, lines 321-324).

4. Please explain the 18% false negative rate for MRI (significant tumors) in Table 2. This is higher than I would have predicted. Is this a result of pathology error? Please explain in more depth how there would be 180 false negatives for significant disease on MRI.

The number of false negatives resulting from the whole mpMRI-based strategy (mpMRI for all then MRI-targeted biopsy if positive) for a cohort of 1000 men is 180. This a result of men with significant prostate cancer missed by mpMRI as well as those men who had a suspicious lesion on mpMRI (mpMRI positive) followed by an erroneous negative MRI-targeted biopsy.

False negatives using MRI-based strategy = (prevalence*1-sensMRI)+(prevalence*sensMRI*1-sensMRITB)*1000pts

$$FN = (0.5*1-0.8)+(0.5*0.8*1-0.8)*1000pts$$

$$FN= 180 pts$$

We have added this explanation into the Results section (Page 14, paragraph 2. lines 253-256). We also provide the calculations in full for these false negatives and for the other diagnostic outcomes in table 2.

5. A problem with basing the assumptions on different published studies is that each study has different disease prevalence rates. For instance, referral hospitals (which often do most of the publishing) may have higher rates (or lower rates) of clinically significant disease compared with a community hospital. This depends highly on how a patient ends up in a certain hospital. It would be preferable to use a collection of single institution data (for TRUS and MRI). The merits of both approaches should be discussed.

An advantage of decision modelling is to draw together different sources of evidence. The sensitivity and specificities of each diagnostic test have been multiplied by the prevalence of clinically significant disease we assume for our base case analysis (50%). Sensitivity and specificity estimates should be independent on prevalence. Of course this is different from carrying out a randomised clinical trial. We have clarified this in the Methods section (page 9, paragraph 1, lines 144-145).

6. I did not follow the reasoning behind changing the MRI specificity from the reported level of 60% to 90%. I agree that the 60% specificity number is incorrect but it seems a bit ad hoc to simply change it to 90%. What happens if you run it at 60%? What about referencing other studies (especially since this was based on cognitive fusions)

The study we used to provide data on MRI-targeted biopsy (not mpMRI) suggested the specificity associated with this test was 80% (not 60%) – see table 1. However the reviewer's point is still valid. We have run a deterministic sensitivity analysis using the sensitivity and specificity estimates reported

in the Kasivisvanathan study (MRI-targeted biopsy sensitivity 80% and specificity 80%) and present this scenario (labelled scenario ii) in the paper.

(See Methods section on page 12, lines 208-209 and 219-222; Results section on page 15, paragraph 2, lines 278-281. Results are also presented in table 3).

7. To avoid confusion on page 11, all of the 4 outcomes should be listed (referenced to Table 2).

Otherwise there is a question about 300 patients not mentioned.

Added to the results section (Page 14, paragraph 1, lines 246-248 and paragraph 2, lines 253-257).

8. Somewhere the term “clinically dominate” should be defined.

This term was explained on what was page 11, paragraph 3, lines 248-251 (in original submitted draft). Now the explanation can be found on page 14, paragraph 3, lines 259-262.

9. Wonder if it would be possible to create an online “calculator” in which other investigator’s data for TRUS and MR guided bx could be entered. If you kept this centrally you could generate worldwide, a lot of data that could generalize across many health care systems and would eliminate patient selection biases by averaging. The input data from the literature is highly selective and could influence the output. The Excel One program might be a simple way to do this.

We don’t think that adding a calculator would add to the value of the paper. The structure of the decision analysis is straightforward and those interested in using it can easily reproduce it within a spreadsheet. To help this process, we have added details of the calculations for each of the results in table 2.

Reviewer 4: Andrew Martin, NHMRC Clinical Trials Centre, University of Sydney

1. Page 3 line 48: Replace "...demonstrates..." with "...suggests..."

Done. (Abstract, now page 4, line 68).

2. Page 10 line 36-41: Report the shape parameters of the beta distributions, or indicate the 2.5% and 97.5% percentile from the distributions specified.

Beta distributions were calculated using the integer form, the parameters are now given in table 1.

3. Page 11 line 7: Replace "...demonstrated..." with "...estimated..."

Done (Results section, now on page 14, paragraph 1, line 242).

4. Page 12 line 17-22: provide details on the results of the 9.6% of simulations where trade-offs arose. We re-ran the analysis and now have slightly different figures due to one small error. The percentage of simulations where trade-offs arose is now 13.2%. We do not believe there are any statistics we could report that would be useful in understanding the trade-offs that occur between identifying true positives and true negatives in these simulations. Instead we have clarified the meaning in the text to better illustrate our point. (Results section, page 16, paragraph 1, lines 286-290).

5. Page 13 line 8: Replace "...revealed..." with "...suggests..."

Done. (Discussion section, now on page 17, paragraph 1, line 294).

6. Page 13 line 12: Replace "...would..." with "...could possibly..."

We have replaced “Indeed, the results show that this MRI-based strategy would reduce the number of biopsies...” with “The results suggest that the MRI-based strategy could reduce the number of biopsies...” (Discussion section, now on page 17, paragraph 1, line 296).

7. Page 14 line 13: The specificity estimate for MRI targeted biopsy was 'bumped up' to match the 90% for TRUS biopsy (page 10 line 3). This would improve the comparative effectiveness of mpMRI rather than underestimate it (as suggested on page 14 line 13).

We think this is an assumption that is clinically valid but we should have tested this with a sensitivity analysis. We now present a sensitivity analysis using a MRI-targeted biopsy specificity of 80% as reported in the Kasivisvanathan paper and report the results. (Methods: page 12, paragraph 1, lines 208-209 and 219-222. Results section, page 15, paragraph 2, lines 278-281. Results are also presented in table 3.).

We also removed the sentence on page 18, lines 319-320 as we felt that it did not add any strength to our argument.

8. Page 16 line 5: Replace "...demonstrates..." with "...suggests..." and "...is likely to..." with "...may..." Done. (Conclusions, page 21, line 366).

VERSION 2 – REVIEW

REVIEWER	Maarten de Roij Radboudumc, Nijmegen, The Netherlands
REVIEW RETURNED	20-May-2014

GENERAL COMMENTS	The authors have updated their manuscript and I have no further comments. All the points are addressed.
---